# Anti-Obesity and Antidiabetic Effects of Nelumbinis Semen Powder in High-Fat Diet-Induced Obese C57BL/6 Mice

**DOI:** 10.3390/nu12113576

**Published:** 2020-11-22

**Authors:** Su Bin Hwang, Bog-Hieu Lee

**Affiliations:** Department of Food and Nutrition, Chung-Ang University, Gyeonggi-do 17546, Korea; subin_0719@nate.com

**Keywords:** Nelumbinis Semen powder, anti-obesity, antidiabetes, high-fat diet-induced obese mice, PPAR-α, PPAR-γ, GLUT4, TNF-α

## Abstract

Nelumbinis Semen (NS, the seeds of *Nelumbo nucifera*) extract is a traditional Korean medicine with anti-oxidant activity. The present study examined the anti-obesity and antidiabetic effects of NS powder in high-fat diet (HFD)-induced obese C57BL/6 mice. Mice (*n* = 8/group) were fed a normal diet (CON), HFD, HFD containing 5% NS powder (HFD-NS5%), or HFD containing 10% NS powder (HFD-NS10%) for 12 weeks. Food intake was relatively higher in groups HFD-NS5% and HFD-NS10%, while the food efficiency ratio was highest in group HFD (*p* < 0.05). HFD-NS5% reduced the body weight (−39.1%) and fat weight (−26.6%), including epididymal fat and perirenal fat, and lowered the serum triglyceride levels (−20.6%) compared with HFD. Groups HFD-NS5% and HFD-NS10% showed hepatoprotective properties, reducing the serum ALT levels (*p* < 0.05) and fat globules (size and number) in the liver compared with group HFD. HFD-NS5% and HFD-NS10% regulated the blood glucose, improved the glucose intolerance, and showed a 12.5% and 15.0% reduction in the area under the curve (AUC) of intraperitoneal glucose tolerance test (IPGTT), and a 26.8% and 47.3% improvement in homeostatic model assessment insulin resistance (HOMA-IR), respectively, compared with HFD (*p* < 0.05). Regarding the expressions of genes related to anti-obesity and antidiabetes, there was a 1.7- and 1.3-fold increase in PPAR-α protein expression, 1.4- and 1.6-fold increase in PPAR-γ protein expression, and 0.7- and 0.6-fold decrease in TNF-α protein expression, respectively, following HFD-NS5% and HFD-NS10% treatments, compared with HFD, and GLUT4 protein expression increased relative to CON (*p* < 0.05). These results comprehensively provide the fundamental data for NS powder’s functional and health-promoting benefits associated with anti-obesity and antidiabetes.

## 1. Introduction

According to the Korea National Health and Nutrition Examination Survey (KNHANES), the prevalence of obesity in the Korean population increased from 29.7% in 2009 to 35.7% in 2018, with the estimated total cost of obesity in Korea estimated to be $1.8 billion [1]. The fundamental cause of obesity is known to be an imbalance between energy intake and expenditure. However, complex causal factors, including dietary choices, genetic predisposition, environmental factors, and the westernization of lifestyles, also play a role [2]. A more scientific description of obesity would be that it results from adipose tissue expansion by adipocyte hypertrophy and hyperplasia, especially in preadipocytes, stroma vascular cells, and stem cells. Two morphologically, anatomically, and physiologically distinct adipose tissue types are brown adipose tissue (BAT) and white adipose tissue (WAT). WAT is the most abundant type found in most organisms and is crucial for energy storage. WAT has also been well-characterized as an endocrine organ that controls a wide range of biological functions. By contrast, BAT acts as a thermal regulator and a protector for important organs [3]. Previous studies showed many health problems associated with obesity and excessive fat, such as dyslipidemia, atherosclerotic diseases, type 2 diabetes mellitus (T2DM), metabolic syndrome, and heart disease [4].

Diabetes is closely related to obesity, a group of metabolic disorders characterized by high blood sugar levels over a prolonged period resulting from either destruction or impairment of insulin-secreting pancreatic cells and insulin action in target tissues. The prevalence of diabetes has increased dramatically in recent decades, claiming 1.3 million mortalities in 2010, a rate around twice as high as that in 1990 [5]. The International Diabetes Federation (IDF) stated that 382 million people worldwide had diabetes in 2013, and this number is expected to rise to 592 million by 2035 [6]. According to KNHANES 2016, 14.4% of Koreans aged over 30 years had diabetes [7]. Diabetes-related complications, such as retinopathy, neuropathy, nephropathy, and cardiovascular and cerebrovascular diseases, are increasing the burden on individuals and healthcare systems. Guidelines for diabetes care recommend controlling glucose, blood pressure, and lipid levels and managing one’s lifestyle (diet, weight control, smoking cessation, and physical activity) to reduce risk factors [8,9,10]. Various strategies have been developed to cure or prevent obesity. Most are based on suppressing appetite, normalizing lipid metabolism (e.g., lipogenesis and lipolysis), and increasing energy expenditure [11].

The fat deposition in WAT, a characteristic of obesity, is regulated by key adipogenic transcription factors responsible for the lipogenesis–lipolysis balance. The lipolysis gene—peroxisome proliferator-activated receptor alpha (PPAR-α)—is abundant in BAT and the liver and regulates fat cells’ growth and differentiation, lipid metabolism, lipoprotein synthesis, and the inflammatory response in tissues. PPAR-α stimulates carnitine palmitoyltransferase (CPT-1), which acts to introduce free fatty acids (FFAs) into the mitochondria and regulate lipid metabolism by promoting β-oxidation of FFAs [12]. Peroxisome proliferator-activated receptor gamma (PPAR-γ) is mainly expressed in adipocytes and regulates lipid and glucose metabolism. It is essential for adipocyte proliferation and differentiation and plays an important role in enhancing anti-inflammation and insulin sensitivity by promoting adiponectin expression in various peripheral tissues [13]. Adiponectin, one of the adipokines, inhibits glucose production in the liver and improves insulin sensitivity, glucose absorption, and fatty acid oxidation in muscles [14]. It has been reported that the synthetic PPAR-γ ligands, thiazolidinediones, increase adiponectin expression and suppress the expression of inflammatory cytokines, such as tumor necrosis factor-α (TNF-α), by activating PPAR-γ in adipocytes [14].

The glucose transporter type 4 (GLUT4) is present in WAT and skeletal muscle, where it regulates the influx of glucose by triggering the insulin signaling pathway [15,16]. The liver is considered an important insulin-sensitive organ, playing key roles in maintaining glucose homeostasis via its physical function of storing and secreting glucose. Hepatic insulin resistance could lead to glycometabolism [15]. Reports have indicated that excessive hepatic gluconeogenesis may largely contribute to hyperglycemia in T2DM [17]. Hence, abnormal hepatic gluconeogenesis might be a therapeutic target for T2DM. TNF-α and interleukin-6 (IL-6), known as inflammatory cytokines, cause chronic low-grade inflammation, which, in turn, may trigger other chronic pathophysiological conditions such as T2DM, metabolic syndrome, and heart disease [4]. Furthermore, TNF-α hampers insulin signaling and stimulates fatty acid lipolysis in adipocytes. Increased FFA flux from adipose tissue to non-adipose tissue reduces the expression of insulin receptor substrate 1 (IRS-1), impairs the activation of phosphoinositide 3-kinase/protein kinase B (PI3K–AKT) signaling in the liver and skeletal muscles, and inhibits GLUT4 translocation to the cell membrane, down-regulating insulin signaling and causing insulin resistance [18]. Insulin resistance means that cells cannot react appropriately to normal insulin levels, leading to hyperinsulinemia [19]. In addition, TNF-α inhibits GLUT4 expression and translocation and inhibits insulin signaling by increasing IRS-1 phosphorylation at inhibitory sites [20]. TNF-α also induces lipolysis, producing circulating lipids, thereby inducing insulin resistance [21]. Insulin resistance and lipid abnormality are closely related, and it has been reported that the triglyceride (TG) level rises, and high-density lipoprotein cholesterol (HDL-C) decreases when insulin resistance increases [20]. Therefore, mediating factors associated with the insulin signaling pathways can be a therapeutic target for improving insulin resistance of T2DM.

Nelumbinis Semen (NS) is the seeds of *Nelumbo nucifera* Gaertn., an aquatic vegetable. NS is widely consumed in Asia (China, Japan, India, and other Southeast Asian countries), the Americas, and Oceania because of its high content of physiologically active substances, including neferine, liensinine, non-crystalline alkaloid N-9, resistant starch, various essential amino acids, alkaloids, anti-oxidant components, and bioactive components [22]. Previous studies have also reported the hepatoprotective effect and other properties, such as anticancer, anti-obesity, and serum lipid improvement of *N. nucifera* Gaertn [23,24,25,26]. It has been reported that the components of each part of *N. nucifera* Gaertn. are similar, but studies on NS are insufficient compared to leaves and roots [23,24]. In addition, most of the studies administered NS to animals in the form of extracts [25,26]. Although the compounds or extracts of NS displayed anti-obesity effects, including reducing fat weight and TG and regulating blood glucose [22,23,24,25,26], the effects of NS in the form of powder in the diet are currently unknown.

Given the close relationship between obesity and diabetes, this study was designed to confirm the comprehensive anti-obesity and antidiabetic effects of NS powder in diet-induced obese C57BL/6 mice for 12 weeks and elucidate the underlying mechanism of these effects. It is anticipated that the data obtained from this study will provide basic support for the functional uses of NS powder.

## 2. Materials and Methods

### 2.1. Anti-Oxidant Activity of NS Extract

NS extract (no. CA02-076, extracted in 95% ethyl alcohol, China) was purchased from the Korea Plant Extract Bank of the Korea Research Institute of Bioscience & Biotechnology (Cheongju, Korea). It was dissolved in dimethylsulfoxide, and the resulting solution was analyzed for anti-oxidant activity by measuring the total polyphenol content, total flavonoid content, and 2,2-diphenyl-1-picrylhydrazyl (DPPH) radical scavenging activity. For the analysis of total polyphenol content, the Folin–Ciocalteu method was used [27]. In a test tube, 40 µL of NS extract dissolved in 50% methanol at 0.2 mg/mL was mixed with 50 µL of Folin–Ciocalteu phenol reagent and then added to 160 µL of 10% sodium carbonate solution after 3 min at room temperature. After incubating the mixture at room temperature for 30 min, the absorbance was measured at 700 nm using a 7315 UV spectrophotometer (Jenway, Staffordshire, UK). The total polyphenol content was expressed as milligrams of gallic acid equivalents (GAE) per gram. To determine the total flavonoid content, 0.5 mL of the methanolic NS extract (0.2 mg/mL) was mixed with 5 mL of diethylene glycol (Duksan Pure Chemicals, Ansan, Korea). After 5 min at room temperature, the mixture was added to 0.5 mL of 1 N NaOH (Duksan Pure Chemicals) and allowed to stand in a 37 °C water bath for 1 h. The absorbance was measured at 420 nm (7315 UV spectrophotometer, Jenway, Staffordshire, UK). The total flavonoid content was expressed as milligrams of naringin (Tokyo Kasei Kogyo Co., Ltd., Tokyo, Japan) equivalents (NE) per gram. The DPPH scavenging activity method was conducted as described by Blois [28], with some modifications. In a test tube, 0.5 mL of NS extract (0.2 mg/mL) was mixed with 3 mL of 0.2 mM methanolic DPPH solution. After 30 min in the dark at room temperature, the absorbance of the mixture was measured at 517 nm (7315 UV spectrophotometer, Jenway). The electron-donating ability was calculated as follows:DPPH radical scavenging activity (%) = (1 − *Abs*/*Abc*) × 100(1)
where *Abs* and *Abc* are the absorbances of DPPH with and without sample at 517 nm, respectively.

### 2.2. Animal Breeding and Experimental Design

Thirty-two 5-week-old male C57BL/6 mice (Dae Han Bio Link Co., Ltd., Eumseong, Korea) were housed under controlled conditions at 23 ± 2 °C, relative humidity 50 ± 5%, and light–dark cycle of 12:12 h. All animal procedures were performed according to the Guide for the Care and Use of Laboratory Animals of the National Institutes of Health and the Animal Welfare Act. Animals were adapted for 1 week with free access to pellet food and water. After adaptation, the mice were randomly assigned to four experimental groups (*n* = 8/group) and raised for 12 weeks. Food intake and body weight (BW) were measured once and twice per week, respectively. The study was approved by the Institutional Animal Care and Use Committee (IACUC) of Chung-Ang University, Seoul, Korea (approval ID: 2019-00006).

### 2.3. Diet Preparation and Formula Composition for Animal Study

The four groups (*n =* 8/group) of mice were fed either a control (CON) diet (NIH-41), high-fat diet (HFD), HFD with 5% NS (HFD-NS5%), or HFD with 10% NS (HFD-NS10%). NIH-41 and the HFD feed (TD.06414) were provided by Dae Han Bio Link Co., Ltd. Table 1 shows the composition of TD.06414. The HFD diet was composed of 60 kcal% fat. The NS powder (79% carbohydrate, 5% fat 16% protein; expiration date 17 December 2020) used in this study was 100% NS without core and outer seed coats, harvested from Vietnam (Jayeonjiae Co., Ltd., Seongnam, Korea).

### 2.4. Blood and Organ Collection

We fed the experimental diets for 12 weeks. On the last day of the experiment, the mice were sacrificed by CO_2_ gas after 12 h fasting, then dissected. Blood was collected via cardiac puncture into heparinized tubes using a syringe and left at room temperature to coagulate for 20 min before centrifugation at 3000× *g*, 5 °C for 15 min. The resultant serum was stored at −70 °C until analysis (Section 2.5). The liver, heart, kidney, spleen, testes, and epididymis (with the epididymal fat, perirenal fat, and BAT) were harvested. Each organ was rinsed in physiological saline, weighed, and stored in a freezer at −70 °C for later use.

### 2.5. Blood Chemicals Analysis

Serum total cholesterol (TC), TG, and HDL-C were determined using commercial enzymatic kits and an automatic biochemistry analyzer (Roche Cobas 8000 modular analyzer Series C702, Mannheim, Germany). Low-density lipoprotein cholesterol (LDL-C) was calculated using the Friedewald formula (TC – HDL − TG/5). The cardiac risk factor (CRF) and atherogenic index (AI) were calculated as CRF = TC/HDL-C and AI = (TC − HDL-C)/HDL, respectively. Alanine aminotransferase (ALT) was analyzed using a Roche Alanine Aminotransferase acc. To IFCC (Mannheim, Germany). Insulin was analyzed using a mouse insulin ELISA kit (Cusabio, Houston, TX, USA). Homeostatic model assessment insulin resistance (HOMA-IR) was calculated using the following formula:HOMA-IR = Fasting insulin (µU/mL) × Fasting blood glucose(mg/dL)/405(2)

### 2.6. Histopathological Analysis

The liver tissues were fixed with 10% neutral formalin. Tissue sections were prepared by means of a routine paraffin embedding process in a histology laboratory. The 5 µm thick tissue slides were stained with hematoxylin and eosin (H&E). A veterinary pathologist examined the tissue for histopathologic lesions.

### 2.7. Blood Glucose Measurement

Blood glucose was measured daily at the same time at 0, 2, 4, 6, 8, and 10 weeks during the experiment period. Following Avertin anesthesia of the mice (10 µL/g BW), blood was collected from the tail vein for measurement of blood glucose using a blood glucose meter (Accu-Check Performa, Roche Diagnostics).

### 2.8. Intraperitoneal Glucose Tolerance Test (IPGTT)

After 12 weeks of experimental feeding, the IPGTT was performed after fasting the mice for 12 h. The mice were injected with 10 μL/g BW of D-glucose at a concentration of 100 mg/mL in saline. Blood was then collected from the tail vein at 0, 30, 60, 90, and 120 min, and blood glucose was measured using a blood glucose meter (Accu-Check Performa, Roche Diagnostics).

### 2.9. Western Blot Analysis

Liver tissues were stored at −70 °C. Tissue samples were homogenized in ice-cold RIPA lysis buffer (Millipore, Billerica, MA, USA) for protein extraction. Tissue debris was removed by centrifugation, and the resulting supernatants were collected and analyzed for protein concentration by the BCA protein assay kit (Thermo Scientific, Bartlesville, OK, USA). The protein was separated on a 10% SDS polyacrylamide gel and then transferred to nitrocellulose membranes (Hybond, GE Healthcare Life Sciences, Little Chalfont, UK). The membranes were incubated with specific primary antibodies overnight at 4 °C. The primary antibodies included anti-mouse β-actin, anti-mouse PPAR-α (Santa Cruz Biotechnology, Santa Cruz, CA, USA), anti-rabbit TNF-α, anti-rabbit PPAR-γ, and anti-rabbit GLUT4 (Cell Signaling Technology, Inc., Beverly, MA, USA). After washing, the membranes were allowed to react with diluted horseradish peroxidase-conjugated secondary antibodies, including goat anti-rabbit IgG antibody and horse anti-mouse IgG (Cell Signaling Technology, Inc., Beverly, MA, USA) at room temperature for 2 h. An enhanced chemiluminescence system (SuperSignal, Thermo Scientific, Rockford, IL, USA) was used to visualize antibody–antigen complexes.

### 2.10. Statistical Analysis

SPSS version 25 (IBM Corp., Armonk, NY, USA) was used for statistical analysis. All experimental data were expressed as mean ± standard error (SE). One-way ANOVA was performed to test whether the means were significantly different among the groups, followed by Duncan’s multiple range test to analyze the differences between pairs of groups. Statistical significance was set at *p* < 0.05.

## 3. Results

### 3.1. Anti-Oxidant Activity of NS Extract

Assessment of the anti-oxidant activities of the 50% methanolic NS extract revealed a total polyphenol content of 49.8 mg GAE/g, a total flavonoid content of 82.8 mg NE/g, and DPPH radical scavenging of 78.8%. According to Kim et al.’s [29] study of the polyphenol contents, flavonoid contents, and anti-oxidant activities of 40 species of Korean natural and medicinal plants, NS had less polyphenol content than the mean value of the 40 species (142.6 mg/g extract). However, its flavonoid content was similar to the high flavonoid content of *Lespedeza cuneata* G. Don (90.2 mg/g extract) and *Artemisia scoparia* Waldst. et Kitamura (77.6 mg/g extract). In addition, among Korean natural plants, *Dryopteris crassirhizoma* and *Cyrtomium fortunei* also showed high DPPH radical scavenging activity (79.5% and 76.5%, respectively) [29], which was similar to our result. The DPPH radical scavenging activity was higher for the NS extract than dibutylhydroxytoluene (74%), used as a preservative. Our results showed that NS possesses high anti-oxidant activity.

### 3.2. Effect of NS Powder on BW Gain, Food Intake, and Food Efficiency Ratio (FER)

Table 2 shows the changes in BW, weight gain, food intake, and FER of mice fed the normal diet (CON), HFD, HFD-NS5%, and HFD-NS10%, respectively, for 12 weeks. There were no significant differences in the initial BW between groups. The final BW increased significantly in mice fed with HFD (39.3 ± 1.7 g), HFD-NS5% (39.1 ± 2.5 g), and HFD-NS10% (41.5 ± 1.5 g) relative to CON (28.8 ± 0.6 g). In addition, weight gain was higher in the HFD groups than in CON (*p* < 0.05). Daily food intake was highest in CON (3.1 ± 0.0 g/day), followed by the HFD-NS10% (3.0 ± 0.1 g/day), HFD-NS5% (2.8 ± 0.1 g/day), and HFD (2.6 ± 0.0 g/day) groups (*p* < 0.05). However, the FER was significantly higher in HFD (8.6 ± 0.7), HFD-NS5% (7.6 ± 0.9), and HFD-NS10% (8.3 ± 0.4) than in CON (2.9 ± 0.2).

### 3.3. Effect of NS Powder on Organ Weight per Unit

Organ weights per unit are presented in Table 3. The liver weight was significantly lower in the HFD groups than in CON. Heart weight was significantly lower in groups HFD-NS5% and HFD-NS10% than in HFD. Kidney weight was significantly lower in HFD compared with CON. Spleen weight was highest in mice fed with HFD (*p* < 0.05). However, epididymal fat was lowest in HFD-NS5% and higher in HFD and HFD-NS10% than in CON (*p* < 0.05). Perirenal fat was higher in the HFD groups than in CON but significantly lower in mice fed with HFD-NS5% relative to HFD and HFD-NS10% among the HFD groups. BAT was highest in mice fed with HFD, and no significant differences were detected among the other three groups.

### 3.4. Effect of NS Powder on Blood Chemical Analysis

The blood chemical analysis results are presented in Table 4. TC and HDL-C were significantly higher in the HFD groups than in CON. In particular, the HDL-C in HFD-NS5% and HFD-NS10% were 1.3- and 1.2-fold higher than HFD, respectively (*p* < 0.05). LDL-C was highest in HFD and HFD-NS10%, but not significantly different between HFD-NS5% and CON. TG was significantly lower in mice fed with HFD-NS5% than with HFD and HFD-NS10%, but there was no significant difference between HFD-NS5% and CON.

AI was significantly lower in HFD-NS5% and HFD-NS10% than in HFD. Although AI did not differ significantly between HFD-NS10% and CON, it was 1.5-fold lower in HFD-NS5% than in CON. In addition, CRF was significantly lower in HFD-NS5% versus HFD and CON, and HFD-NS5% versus HFD. Among all four groups, the liver damage indicator, ALT, was highest in HFD, with no significant differences among the remaining three groups. Serum insulin concentration (Figure 1) was significantly higher in HFD than in CON. Of the HFD groups, the serum insulin concentrations in mice fed with HFD-NS5% and HFD-NS10% were both significantly lower than in mice fed with HFD. HOMA-IR, an insulin resistance indicator, was 26.9% lower in HFD-NS5% and 47.3% lower in HFD-NS10% relative to HFD, but HFD-NS5% and HFD-NS10% were not significantly different from CON (Figure 1).

### 3.5. Effect of NS Powder on Histopathological Changes in the Liver

Figure 2 shows the H&E staining of the liver. The percentage of normal liver cells was 100% in CON, 50% in HFD-NS5% and HFD-NS10%, separately, and just 12.5% in HFD. In addition, histologically, HFD showed a notable increase in fat vacuoles when compared with CON. Both HFD-NS5% and HFD-NS10% showed dramatic decreases in fat vacuoles (both size and number) relative to HFD.

### 3.6. Effect of NS Powder on Random Blood Glucose Levels

Figure 3 shows the results of random blood glucose tests taken at consistent times during weeks 0, 2, 4, 6, 8, and 10. There were no significant differences among all four groups at week 0. HFD-NS10% maintained significantly lower blood glucose than CON at weeks 6 and 8. However, at week 10, HFD (165.6 ± 2.9 mg/dL), HFD-NS5% (149.0 ± 5.8 mg/dL), and HFD-NS10% (147.3 ± 4.8 mg/dL) recorded higher blood glucose contents relative to CON (133.3 ± 3.2 mg/dL) (*p* < 0.05), but the blood glucose levels for HFD-NS5% and HFD-NS10% remained lower compared with HFD.

In addition, the mean blood glucose levels in groups HFD-NS5% and HFD-NS10% were lower, respectively, by 10% and 11% relative to group HFD, results similar to or lower than those in group CON at weeks 0–10 (Table 5). Weekly blood glucose decreased from week 4 in CON, HFD, and HFD-NS10%, while in HFD-NS5% it decreased from week 2 (*p* < 0.05).

### 3.7. Effect of NS Powder on IPGTT

After 12 h fasting, IPGTT was performed. The results are presented in Figure 4. Fasting blood glucose, which was measured in each group before D-glucose was injected, was found to be significantly higher in the HFD groups, although HFD-NS5% (114.3 ± 2.5 mg/dL) and HFD-NS10% (122.2 ± 3.3 mg/dL) were both lower than HFD (134.4 ± 3.6 mg/dL) at 0 min (*p* < 0.05). At 30 min, HFD-NS5% and HFD-NS10% had similar to or lower blood glucose levels, respectively, compared with CON, and both were significantly lower relative to HFD. In particular, the HFD-NS10% result was 1.3-fold lower than that of HFD. At 60 and 90 min, HFD-NS5% and HFD-NS10% were significantly lower than HFD, but not differing significantly from CON. At 120 min, HFD (186.4 ± 4.4 mg/dL) was higher than CON (156.5 ± 3.1 mg/dL) (*p* < 0.05), whereas HFD-NS5% (172.1 ± 14.2 mg/dL) and HFD-NS10% (165.0 ± 0.9 mg/dL) were similar to CON. Figure 4 shows that the area under the curve of IPGTT was, respectively, 12.5% and 15.0% lower in HFD-NS5% and HFD-NS10% than in HFD, supporting that NS supplementation reduces glucose intolerance.

### 3.8. Effect of NS Powder on PPAR-α, PPAR-γ, GLUT4, and TNF-α Protein Expression

In the liver, PPAR-α protein expression was higher in groups HFD-NS5% and HFD-NS10% than in HFD. Notably, PPAR-α expression in HFD-NS5% increased more than in the CON group (*p* < 0.05) (Figure 5A). PPAR-γ protein expression was lowest in HFD, whereas groups HFD-NS5% and HFD-NS10% were 1.4- and 1.6-fold higher than HFD, respectively (Figure 5B). In addition, GLUT4 protein expression was lowest in HFD, and higher in groups HFD-NS5% and HFD-NS10% than in HFD, indeed more than in CON (*p* < 0.05) (Figure 5C). TNF-α protein expression was lower in HFD-NS5% and HFD-NS10% than in HFD. In particular, HFD-NS10% decreased TNF-α protein expression more than the CON group (*p* < 0.05), while HFD-NS5% and CON were statistically similar (Figure 5D).

## 4. Discussion

In this study, we assessed the anti-oxidant activities of NS extract and the anti-obesity and antidiabetic properties of dietary supplementation with 5% and 10% NS powder in HFD-induced obese C57BL/6 mice for 12 weeks. An initial evaluation of the anti-oxidant properties of the NS powder revealed a low total polyphenol content compared with 40 species of Korean natural and medicinal plants [29]. By contrast, the flavonoid content and DPPH radical scavenging activity were relatively high. Flavonoids possess anti-oxidant, anti-virus, anti-inflammatory, anticancer, and cardioprotective activities, besides displaying hypoglycemic effects [22]. NS contains various phytochemicals, such as quercetin, kaempferol 3-*O*-robinobioside, rutin, catechin, isoquercitrin, astragalin, and hydroxybenzoic acid [30]. Of these flavonoids, quercetin, catechin, astragalin, isoquercitrin, and hydroxybenzoic acid have lipolytic activities in visceral fat [31]. Rutin is also reported to be effective in inhibiting adipocytes differentiation [32]. Quercetin is a potent inhibitor of α-glucosidase activities, more so than acarbose. α-Glucosidase breaks down starch so that it can be absorbed in the small intestine [33]. Thus, inhibiting this enzyme can alleviate hyperglycemia after a meal [34]. Although quantitative analysis of the individual flavonoid components was not performed in this study, the strong anti-oxidant activity of the NS extract was shown to be attributed to the relatively high total flavonoid content and DPPH scavenging activity.

To determine the anti-obesity effects of NS powder, we fed mice with HFD containing 5% and 10% NS powder for 12 weeks. BW increased in all HFD groups as compared with the CON group. Food intake was higher in mice fed with HFD-NS5% and HFD-NS10% than in HFD-fed mice (*p* < 0.05). However, FER was the highest in HFD-fed mice and the lowest in mice fed HFD-NS5% (*p* < 0.05). High FER means that weight gain is high even if eating less. Compared with the HFD group, the food intakes of groups HFD-NS5% and HFD-NS10% were higher, but FER was lower (*p* < 0.05), so we can conclude that the anti-obesity effects of NS powder are not related to appetite suppression. In this study, the HFD-NS5% group exhibited a significant decrease in the epididymal fat weight per unit (−39.1%) versus the HFD group, and the perirenal fat weight per unit also decreased (−28.6%), which was lower than that in both the HFD and CON groups (*p* < 0.05).

In one study, oral administration of the *N. nucifera* ethanol extract (220 mg/kg BW/day) for 30 days inhibited adipogenesis, lowered fat weight, and improved the lipid profiles in HFD-fed rats [35]. In another study, an oral administration of lotus leaf ethanol extract (1.22 g/kg) for 5 weeks promoted lipolysis in WAT in HFD-induced obese mice [31]. In addition, *N. nucifera* alkaloids, a major component of lotus leaf, are reported to induce apoptosis in 3T3-L1 preadipocytes [35]. You et al. [25] administered ethanol extracts of NS (400 mg/kg) orally to HFD-induced obese Sprague-Dawley rats for 7 weeks. The epididymal fat and abdominal visceral fat of the treated groups decreased by up to 19.2%, compared with HFD alone.

In the current study, PPAR-α expression in HFD-NS5% increased by 1.7 times, and HFD-NS10% increased by 1.3 times compared to HFD. According to Cho et al., an experimental mice group to which chlorogenic acid (0.2 g/kg diet) was administered for 8 weeks also decreased epididymal fat and perirenal fat weight, while PPAR-α expression increased, as compared to obese mice induced to obesity by an HFD (37 kcal% from fat) [36]. Some studies have shown that HFD reduced the expression of PPAR-α and carnitine palmitoyl transferase-1 (CPT-1, an energy expenditure gene) in the WAT of obese humans and animals [36,37]. PPAR-α, a lipolytic gene, existing abundantly in BAT, liver, and to a lesser extent in kidney, heart, and muscle, mainly plays a role in the growth and differentiation of adipocytes, lipid metabolism, and lipoprotein synthesis, and in regulating the inflammatory response in tissues [38]. PPAR-α stimulates CPT-1, which acts to introduce FFAs into mitochondria and regulate lipid metabolism through promoting and utilizing the energy derived from β-oxidation of FFA [12,37].

In our study, the high-dose HFD-NS10% anti-obesity effects were not clear, but in the low-dose HFD-NS5% group, both epididymal fat and perirenal fat weight decreased. Additionally, the liver staining results showed that the size and number of adipocytes in both the HFD-NS5% and HFD-NS10% groups were improved. It is thought that the NS powder increases PPAR-α expression, which, in turn, promotes the β-oxidation of FFA, the net result being a reduction in body fat accumulation and improvement on lipid profiles.

Kim et al., who studied the effects of powdered lotus leaves, lotus stems, and lotus pods (10% weight diet, respectively) on blood biochemistry in HFD-induced obese mice for 60 days, reported that the serum TG level decreased by 33.1% in the powdered lotus pods group as compared to mice fed with HFD [23]. In another study [25], oral administration of lotus seed ethanol extracts (400 mg/kg/day) reduced the TG levels by 39.1% compared with HFD. In the present study, HFD-NS5% decreased serum TG by 20.6% compared to HFD. In addition, HFD-NS5% was 25.8% higher in HDL-C and 38.3% lower in LDL-C as compared to HFD, while HFD-NS10% was 17% higher in HDL-C as compared to HFD (*p* < 0.05). The extracts of lotus root and lotus leaf have been shown to reduce serum TG, TC, and LDL-C levels [22]. Abundant flavonoids in lotus leaf reduced TC, TG, and LDL-C levels and increased HDL-C level, which is thought to improve lipid profiles [34]. Additionally, AI (an index of arteriosclerosis) and CRF (an index of the risk of cardiovascular diseases) were lower in both the 5% and 10% NS powder groups when compared to HFD. Notably, the HFD-NS5% value was lower than the CON value (*p* < 0.05). High LDL-C levels and low HDL-C levels are strongly associated with the risk of cardiovascular diseases, while dyslipidemia refers to increased lipids (e.g., TC and TG) in the blood [39]. Therefore, the NS powder may affect serum TG and cholesterol levels and lipid profiles and reduce the risk of chronic diseases, such as dyslipidemia, arteriosclerosis, and cardiovascular diseases, associated with T2DM.

This study revealed that HFD-NS10% showed weak effects on anti-obesity and improved lipid profiles compared to HFD-NS5%. These results are similar to a previous study in which noni juice (50 and 100 mg/kg BW) was orally administered to rats subjected to induced dyslipidemia by HFD. In that study, the weight reduction efficacy of NS powder was not dose-dependent. In addition, the high-dose group did not show any effect on LDL-C but did improve lipid profiles, increasing HDL-C and decreasing TG compared to HFD [40]. Deleterious effects have been reported on the interaction between phenolic components and proteins, lipids, and carbohydrates. For instance, proanthocyanidins and monomeric flavonoids impact starch digestibility through structural modifications of starch when present at high levels (above 10% on a starch weight basis) [41]. The addition of polyphenol to bread dough causes a marked change in the textural properties of bread because of non-covalent cross-linking between wheat protein, starch, and pectin [41]. Other studies show that phenolic–protein interactions induce the aggregation and precipitation of proteins and reduce the bioavailability of flavonoids [41]. Protein–lipid–phenolic and protein–phenolic interactions in flaxseeds and soybeans are reported to reduce the total polyphenol contents and anti-oxidant activity [22]. Therefore, the weak effects of HFD-NS10% on anti-obesity and blood lipid improvement might be explained by an increase in unknown interactions involving the various components in the NS powder, present in excess. These interactions are presumed to reduce the positive effects of the NS powder on anti-obesity and lipid improvement.

Analysis of the liver enzyme ALT activity as an indicator of liver damage revealed lower ALT levels in both the HFD-NS5% and HFD-NS10% groups than in group HFD (*p* < 0.05). According to a previous study, extracts of *N. nucifera* seeds inhibited CCl_4_-induced cytotoxicity and dose-dependently inhibited aflatoxin B_1_ genotoxicity in a rat hepatocyte model [24]. These results support the H&E staining results of the liver in our study, which revealed that the proportion of normal cells in the groups supplemented with NS powder increased compared to HFD without NS, suggesting that the NS powder has hepatoprotective effects.

Based on the random blood glucose measurements, the blood glucose was higher in HFD than CON during the experimental period. Conversely, HFD-NS5% and HFD-NS10% had significantly lower levels than the HFD-fed mice. In a previous study, the blood glucose decreased to within the normal range in streptozotocin- and glucose-induced diabetic rats treated with methanol extracts of lotus root and stem (300 and 600 mg/kg) and tryptophan extracted from the nodes of rhizome (100 and 400 mg/kg) [42]. Neferine (150 µM), an alkaloid component of NS, stimulated an increase in intracellular Ca^2+^ levels, subsequently promoting the binding of the GLUT4 protein to the plasma membrane, which increased glucose uptake in insulin-sensitive L6 myoblasts cells [43]. Administration of neferine (5 mg/kg) by gastrogavage for 3 weeks effectively suppressed hyperglycemia in high-fat- and high-sucrose-induced rats [44]. Nuciferine and norcoclaurine, components of lotus seed, effectively alleviated blood glucose in alloxan-induced diabetic rats [26]. Administration of alloxan destroys β-cells in the islets of Langerhans, causing insulin deficiency and increasing blood glucose [26]. In addition, 70% ethanolic *Lycium chinense* Mill extracts (0.25, 0.5, 1, and 2 mg/mL) known to contain rutin, one of the components of NS, increased GLUT4 expression in 3T3-L1 cells and lowered blood glucose [45]. The citrus flavonoid, nobiletin (100 mg/kg, oral administration for 5 weeks), improved hyperglycemia and insulin resistance by increasing GLUT4 translocation in WAT and muscles of HFD-induced obese mice [46]. In the present study, we also found that the expression of GLUT4 protein increased with NS supplementation. This increase in GLUT4 protein expression might act to improve blood glucose levels.

To determine whether glucose was translocated from the blood into the cells, we conducted an IPGTT. After glucose administration, the blood glucose level was significantly lower in the groups supplemented with NS powder relative to group HFD. This alleviation of the increased blood glucose by NS powder might be due to glucose intake being within the normal range, contributing to alleviating hyperglycemia and glucose intolerance after meals. In addition, NS is a high-amylose food containing about 40% amylose starch and is reported to be involved in blood glucose control [47]. It is known that the high amylose content is a source of resistant starch, and NS is composed of about 45.8% resistant starch. Resistant starch plays an important role in maintaining normal blood glucose range because it is not digested [48,49]. From these findings, it is thought that the effects of lowering the blood glucose and alleviating postprandial hyperglycemia derive, at least in part, from the high amylose and high resistant starch contents and various anti-oxidant compounds in the NS powder.

After 12 weeks of the NS powder-containing diet, serum insulin and HOMA-IR decreased significantly in groups HFD-NS5% and HFD-NS10% compared to group HFD. Kim et al. reported that lotus stem and lotus pods decreased serum insulin levels [23]. It is known that normal insulin action is activated by insulin binding to the insulin receptor of the membrane, followed by tyrosine phosphorylation of IRS-1 and IRS-2, leading to the next metabolism step [50]. Insulin resistance implies impairment of glucose and lipid metabolism and is closely related to obesity. Insulin resistance may occur due to a primary deficiency of IRS, but this is rare. Most instances are caused by impaired signaling pathways after the binding of insulin with its receptor. The most important mechanism of insulin resistance is that the increased FFA levels inhibit tyrosine phosphorylation of IRS-1, thereby inducing inactivation and inhibiting GLUT4 translocation to the cell surface, down-regulating insulin signaling [51]. In addition, insulin resistance is closely associated with lipid abnormalities; thus, as the insulin resistance increases, TG increases and HFD-C decreases [52].

PPAR-γ, mainly expressed in adipocytes, regulates gene expression involved in adipocyte differentiation, regulates lipid and glucose metabolism in WAT [14], and plays important roles in enhancing insulin sensitivity in several peripheral tissues. Its anti-inflammatory and insulin sensitivity improving effects occur via the induction of adiponectin [13]. Increased FFA levels due to lipid metabolic disorders are known to induce inflammation associated with insulin resistance and T2DM mechanism, reducing glucose uptake, increasing hepatic glucose production, impairing insulin signaling pathways, and causing adipokine secretion disorders [16,53]. Adiponectin, one of the adipokines, inhibits hepatic glucose production, lessening insulin resistance by increasing glucose uptake and fatty acid oxidation in muscles [54]. Synthetic PPAR-γ ligands (thiazolidinediones) have been shown to increase adiponectin expression and suppress the expression of inflammatory cytokines, such as TNF-α, by activating PPAR-γ in adipocytes [14]. It is thought that the NS powder increases PPAR-γ expression, which, in turn, enhances insulin sensitivity.

Conversely, inflammatory cytokines, such as TNF-α, IL-6, and monocyte chemoattractant protein-1 (MCP-1), impair insulin signaling pathways targeting the downstream components of the insulin signaling pathway in adipose tissue. Inflammatory cytokines activate the inflammatory signaling pathways, including the inhibitor of nuclear factor-kB kinase B (IKKB) (IKK/NF-kB) and c-Jun NH_2_-terminal kinase (JNK) pathways, and inhibit insulin signaling pathways by regulating IRS-1 serine phosphorylation [4,19]. TNF-α and IL-6 interrupt GLUT4 expression and translocation and interfere with insulin signaling by increasing IRS-1 phosphorylation at inhibitory sites [20]. TNF-α inhibits adipocytes differentiation in human preadipocytes and 3T3-L1 cells and impairs adipokine secretion. In addition, TNF-α induces lipolysis, generating circulation lipids and inducing insulin resistance [16,21]. TNF/TNF receptor knockout mice have been reported to have improved insulin sensitivity [55]. Administration of neferine (5 mg/kg) for 3 weeks via gastrogavage has been reported to decrease fasting blood glucose, insulin, TG, and TNF-α and increase insulin sensitivity in insulin resistance-induced rats [44]. In the present study, the mice that supplemented the NS powder showed lower TNF-α protein expression than in mice fed with HFD (*p* < 0.05). This decrease in TNF-α protein expression might act to reduce inflammation and insulin resistance.

Given these findings, we confirmed that NS powder was effective for reducing fat weight and improving lipid profiles through promoting PPAR-α expression; improving insulin sensitivity through promoting PPAR-γ expression; reducing inflammation through reducing TNF-α expression, besides regulating blood glucose and alleviating glucose intolerance through promoting GLUT4 expression, which is also known to be inhibited by TNF-α. Consequently, we confirmed that NS powder had antidiabetic effects, such as alleviating blood glucose and reducing insulin resistance, besides anti-obesity effects.

This study has several limitations. To determine how much the NS powder supplementation alleviated the characteristics induced by the HFD and approached the level of normal mice, we fed the CON group with a normal mice diet rather than a low-fat control diet or a positive control diet. In addition, we could not measure the body composition, one of the anti-obesity indicators, and additional variables because of the laboratory limitations. Therefore, further studies are needed to decipher the in-depth mechanisms underlying the observed effects. Notwithstanding these limitations, the present study has several strengths, including its long-term experimental period. Most previous studies concern other parts of *N. nucifera* Gaertn., such as leaves and roots [23,24] or as extracts [25,26], and no prior studies evaluated both the anti-obesity and antidiabetes effects. In contrast, we confirmed the effect of NS powder in the diet. In addition, we confirmed the comprehensive properties of NS powder related to anti-obesity and antidiabetes.

## 5. Conclusions

In this study, we determined the anti-obesity and antidiabetic effects of NS powder in HFD-induced obese mice. The NS extract showed high anti-oxidant activities. NS powder supplementation significantly reduced the epididymal and perirenal fat weights and improved the serum lipid profiles, such as lowering serum TC and TG levels. HFD-NS10% did not show distinct anti-obesity effects on BW and fat weight, but HFD-NS5% and HFD-NS10% improved the lipid profile, such as HDL-C, AI, and CRF. Histopathological staining of the liver confirmed that the increased size and number of fat vacuoles due to the HFD were decreased in the group fed with NS powder. Confirmation of the hepatoprotective effects of NS powder supplementation, evidenced by the lower ALT levels, an indicator of liver damage, was observed compared to mice fed with HFD alone. During the 12-week treatment, random blood glucose levels and IPGTT were measured. Our results confirmed that the supplementation with NS powder alleviated hyperglycemic and glucose intolerance induced by the HFD. The HOMA-IR level, an insulin resistance index, also showed a lower value in the NS powder supplementation groups than in group HFD, thereby confirming the insulin resistance-lowering effect of NS powder. The PPAR-α protein expression increased in groups HFD-NS5% and HFD-NS10% compared to group HFD (*p* < 0.05). It is considered that NS powder promoted lipolysis and demonstrated anti-obesity effects. HFD-NS5% and HFD-NS10% increased PPAR-γ protein expression, and GLUT4 protein expression was increased even more than in group CON. It seems that the results show an impact on improving insulin resistance and blood regulation by glucose uptake. Furthermore, the NS powder supplementation decreased the increased TNF-α protein expression levels brought on by the HFD. Thus, the NS powder is considered effective in alleviating the interruption of insulin signaling pathways, inhibiting lipolysis, and improving insulin resistance. Therefore, these results suggest that the NS powder has positive anti-oxidant, anti-obesity, and antidiabetic effects and could be used for functional and health-promoting activities, such as reducing obesity, T2DM, and insulin resistance. However, because the efficacy of NS powder at different concentrations did not show a clear tendency in the individual anti-obesity properties, further research should be undertaken.

## Figures and Tables

**Figure 1 nutrients-12-03576-f001:**
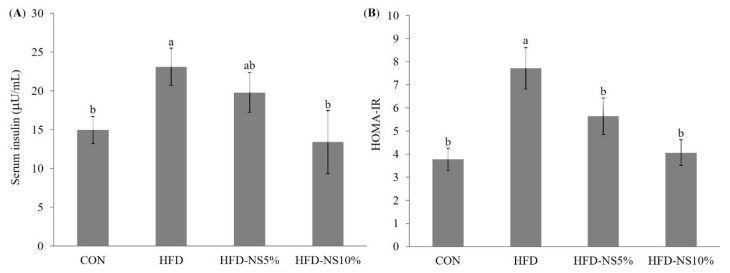
Effects of Nelumbinis Semen powder on insulin resistance in C57BL/6 mice fed a normal diet (CON), high-fat diet (HFD), high-fat diet with 5% Nelumbinis Semen powder (HFD-NS5%), and high-fat diet with 10% Nelumbinis Semen powder (HFD-NS10%). (**A**) Effect of Nelumbinis Semen powder on serum insulin level and (**B**) homeostatic model assessment insulin resistance (HOMA-IR). Values are the mean ± SE (*n* = 8/group). ^a,b^ The different letters represent significant differences at *p* < 0.05 by Duncan’s multiple range test.

**Figure 2 nutrients-12-03576-f002:**
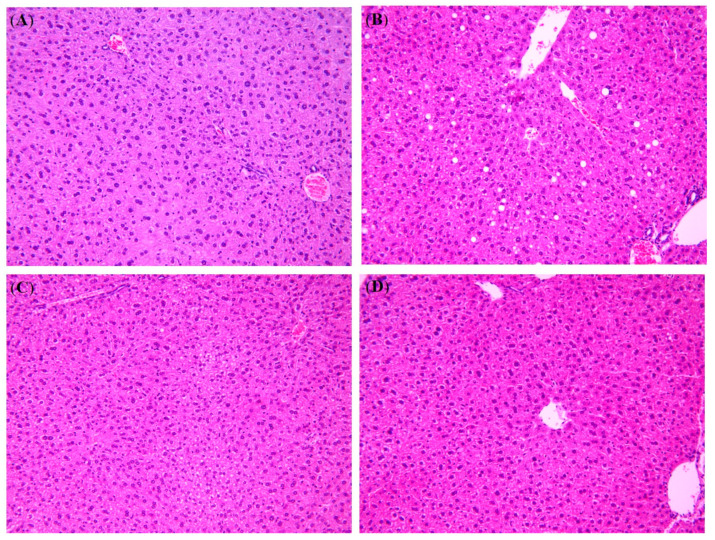
Effect of Nelumbinis Semen powder on histopathological changes in the liver of C57BL/6 mice administered a high-fat diet with low- or high-dose Nelumbinis Semen powder (5% and 10%, respectively). (**A**) Control group (CON), (**B**) high-fat diet (HFD), (**C**) high-fat diet with 5% Nelumbinis Semen powder (HFD-NS5%), and (**D**) high-fat diet with 10% Nelumbinis Semen powder (HFD-NS10%). Hematoxylin and eosin staining showing the fat distribution of hepatocytes (200× magnification).

**Figure 3 nutrients-12-03576-f003:**
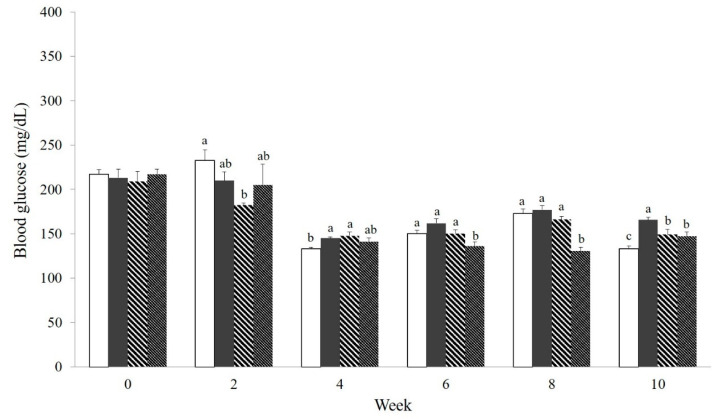
Effect of Nelumbinis Semen powder on random blood glucose levels at weeks 0, 2, 4, 6, 8, and 10 in C57BL/6 mice. 
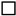
, CON, control group; 
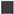
, HFD, high-fat diet; 
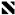
, HFD-NS5%, high-fat diet with 5% Nelumbinis Semen powder; 
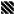
, HFD-NS10%, high-fat diet with 10% Nelumbinis Semen powder. Values are the mean ± SE (*n* = 8). ^a–c^ The different letters represent significant differences at *p* < 0.05 by Duncan’s multiple range test.

**Figure 4 nutrients-12-03576-f004:**
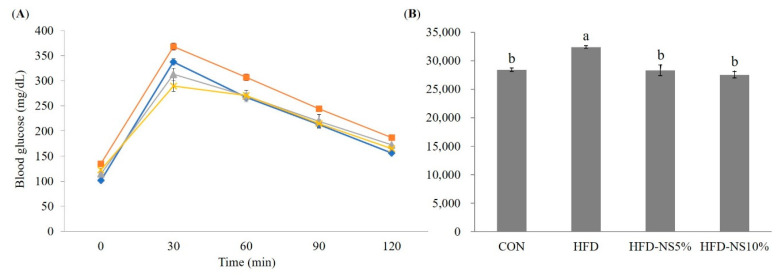
Effect of Nelumbinis Semen powder, normal diet, and high-fat diet on intraperitoneal glucose tolerance test (IPGTT) in C57BL/6 mice administered a high-fat diet with low- or high-dose Nelumbinis Semen powder (5% and 10%, respectively). (**A**) Effect of Nelumbinis Semen powder on IPGTT 
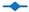
, CON, control group; 
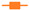
, HFD, high-fat diet; 
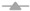
, HFD-NS5%, high-fat diet with 5% Nelumbinis Semen powder; 
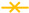
, HFD-NS10%, high-fat diet with 10% Nelumbinis Semen powder. (**B**) Effect of Nelumbinis Semen powder on the area under the curve of IPGTT. Values are the mean ± SE (*n* = 8/group). ^a,b^ The different letters represent significant differences at *p* < 0.05 by Duncan’s multiple range test.

**Figure 5 nutrients-12-03576-f005:**
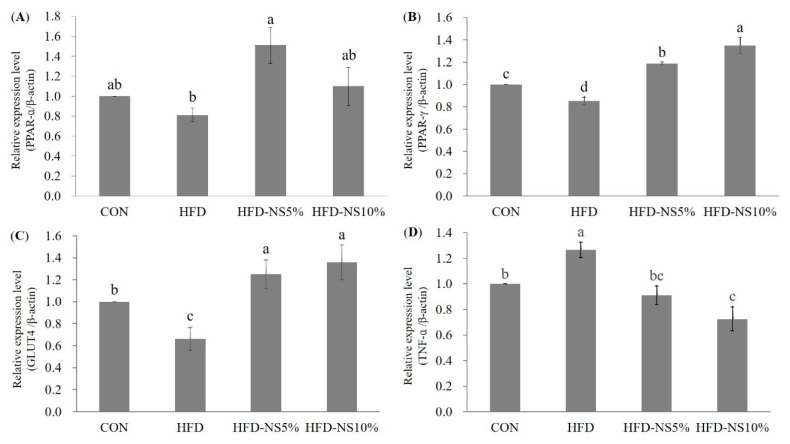
Effects of Nelumbinis Semen powder on protein expression in C57BL/6 mice fed a normal diet (CON), high-fat diet (HFD), high-fat diet with 5% Nelumbinis Semen powder (HFD-NS5%), and high-fat diet with 10% Nelumbinis Semen powder (HFD-NS10%) (*n* = 8/group). (**A**) Relative expression level of PPAR-α/β-actin. (**B**) Relative expression level of PPAR-γ/β-actin. (**C**) Relative expression level of GLUT4/β-actin. (**D**) Relative expression level of TNF-α/β-actin. ^a–d^ The different letters represent significant differences at *p* < 0.05 by Duncan’s multiple range test.

**Table 1 nutrients-12-03576-t001:** Composition (% by weight) of the HFD formula.

Composition	Group ^1^		
	HFD	HFD-NS5%	HFD-NS10%
Casein	26.50	25.18	23.85
L-Cysteine	0.40	0.38	0.36
Maltodextrin	16.0	15.2	14.4
Sucrose	9.00	8.55	8.10
Lard	31.00	29.45	27.90
Soybean oil	3.00	2.85	2.70
Cellulose	6.55	6.55	5.90
Mineral mix ^2^	4.80	4.56	4.32
Calcium phosphate	0.34	0.32	0.31
Vitamin mix ^3^	2.10	2.00	1.89
Choline bitartrate	0.30	0.29	0.27
Blue food color	0.01	0.01	0.01
Nelumbinis Semen powder ^4^	-	5	10

^1^ HFD, high-fat diet; HFD-NS5%, high-fat diet +5% Nelumbinis Semen powder; HFD+NS10%, high-fat diet +10% Nelumbinis Semen powder. ^2^ Mineral mixture according to AIN-93G-MX. ^3^ Vitamin mixture according to AIN-93-VX. ^4^ Nelumbinis Semen powder (% by dry weight).

**Table 2 nutrients-12-03576-t002:** Effect of Nelumbinis Semen powder on body weight gain, food intake, and FER (*n* = 8/group).

	Group ^1^			
	CON	HFD	HFD-NS5%	HFD-NS10%
Initial body weight (g)	21.45 ± 0.21 ^NS^	20.68 ± 0.45	20.96 ± 0.55	20.73 ± 0.20
Final body weight (g)	28.80 ± 0.60 ^b^	39.27 ± 1.67 ^a^	39.05 ± 2.49 ^a^	41.45 ± 1.51 ^a^
Body weight gain (g/week)	0.61 ± 0.04 ^b^	1.55 ± 0.13 ^a^	1.51 ± 0.19 ^a^	1.73 ± 1.12 ^a^
Total food intake (g)	256.43 ± 2.17 ^a^	215.46 ± 3.83 ^c^	235.63 ± 5.03 ^b^	249.46 ± 5.35 ^a^
Daily food intake (g/day)	3.08 ± 0.02 ^a^	2.58 ± 0.04 ^c^	2.83 ± 0.06 ^b^	2.99 ± 0.07 ^a^
FER	2.87 ± 0.19 ^b^	8.59 ± 0.67 ^a^	7.63 ± 0.90 ^a^	8.26 ± 0.43 ^a^

^1^ CON, control group; HFD, high-fat diet; HFD-NS5%, high-fat diet +5% Nelumbinis Semen powder; HFD-NS10%, high-fat diet +10% Nelumbinis Semen powder. Data expressed as mean ± SE. ^a–c^ Values with different superscripts in the same row are significantly different at *p* < 0.05 by Duncan’s multiple range test. FER (food efficiency ratio) = body weight gain/food intake per week; NS: not significant.

**Table 3 nutrients-12-03576-t003:** Effect of Nelumbinis Semen powder on unit organ weight (*n* = 8/group).

		Group ^1^			
		CON	HFD	HFD-NS5%	HFD-NS10%
Organ weight	Liver	35.35 ± 0.33 ^a^	28.52 ± 0.17 ^b^	25.38 ± 0.31 ^c^	24.39 ± 0.30 ^d^
	Heart	3.64 ± 0.08 ^a^	3.55 ± 0.06 ^a^	3.17 ± 0.09 ^b^	2.87 ± 0.05 ^c^
	Kidney	11.36 ± 0.18 ^a^	10.02 ± 0.15 ^b^	9.46 ± 0.14 ^c^	8.38 ± 0.06 ^d^
	Spleen	1.69 ± 0.04 ^b^	1.80 ± 0.03 ^a^	1.41 ± 0.05 ^d^	1.51 ± 0.02 ^c^
	Testis	6.90 ± 0.18 ^a^	5.83 ± 0.11 ^b^	4.74 ± 0.08 ^c^	4.43 ± 0.03 ^c^
	Epididymis	0.70 ± 0.01 ^a^	0.60 ± 0.02 ^b^	0.45 ± 0.01 ^c^	0.47 ± 0.01 ^c^
Fat weight	Epididymal fat	0.70 ± 0.02 ^b^	0.92 ± 0.04 ^a^	0.56 ± 0.04 ^c^	0.85 ± 0.03 ^a^
	Perirenal fat	25.12 ± 0.79 ^c^	59.94 ± 1.12 ^a^	42.82 ± 2.13 ^b^	56.98 ± 0.64 ^a^
	Brown fat	2.48 ± 0.09 ^b^	2.85 ± 0.09 ^a^	2.43 ± 0.02 ^b^	2.27 ± 0.08 ^b^

^1^ CON, control group; HFD, high-fat diet; HFD-NS5%, high-fat diet +5% Nelumbinis Semen powder; HFD-NS10%, high-fat diet +10% Nelumbinis Semen powder. Data expressed as mean ± SE (mg/g body weight). ^a–d^ Values with different superscripts in the same row are significantly different at *p* < 0.05 by Duncan’s multiple range test. NS: not significant.

**Table 4 nutrients-12-03576-t004:** Effect of Nelumbinis Semen powder on blood chemical analysis (*n* = 8/group).

	Group ^1^			
	CON	HFD	HFD-NS5%	HFD-NS10%
Total cholesterol (mg/dL)	59.20 ± 3.43 ^b^	81.90 ± 6.37 ^a^	89.95 ± 3.62 ^a^	90.50 ± 3.22 ^a^
LDL cholesterol (mg/dL)	6.25 ± 0.31 ^b^	11.75 ± 0.41 ^a^	7.25 ± 0.56 ^b^	10.50 ± 1.21 ^a^
HDL cholesterol (mg/dL)	44.25 ± 2.96 ^c^	58.25 ± 5.78 ^b^	73.25 ± 3.05 ^a^	68.75 ± 1.93 ^a,b^
Triglyceride (mg/dL)	43.50 ± 1.02 ^b^	59.50 ± 1.38 ^a^	47.25 ± 3.12 ^b^	56.25 ± 0.56 ^a^
AI (mg/dL)	0.34 ± 0.01 ^b^	0.43 ± 0.03 ^a^	0.23 ± 0.01 ^c^	0.32 ± 0.01 ^b^
CRF (mg/dL)	1.34 ± 0.01 ^b^	1.43 ± 0.03 ^a^	1.23 ± 0.01 ^c^	1.32 ± 0.01 ^b^
ALT (U/L)	78.75 ± 4.93 ^b^	135.50 ± 14.05 ^a^	63.75 ± 2.66 ^b^	79.00 ± 5.10 ^b^

^1^ CON, control group; HFD, high-fat diet; HFD-NS5%, high-fat diet +5% Nelumbinis Semen powder; HFD-NS10%, high-fat diet +10% Nelumbinis Semen powder. Data expressed as mean ± SE. ^a–c^ Values with different superscripts in the same row are significantly different at *p* < 0.05 by Duncan’s multiple range test. NS: not significant. LDL: low-density lipoprotein; HDL: high-density lipoprotein; AI: atherogenic index = (TC − HDL)/HDL; CRF: cardiac risk factor = TC/HDL; ALT: alanine aminotransferase.

**Table 5 nutrients-12-03576-t005:** Effect of Nelumbinis Semen powder on random blood glucose levels (mg/dL) at weeks 0, 2, 4, 6, 8, and 10 (*n* = 8/group).

	Group ^1^			
	CON	HFD	HFD-NS5%	HFD-NS10%
Week 0	216.88 ± 5.19 ^NS,A^	213.13 ± 9.46 ^A^	209.14 ± 11.24 ^A^	217.20 ± 5.45 ^A^
Week 2	232.57 ± 11.63 ^a,A^	209.80 ± 9.89 ^a,b,A^	182.20 ± 2.17 ^b,B^	205.13 ± 23.07 ^a,b,A^
Week 4	133.00 ± 1.95 ^b,C^	145.00 ± 1.56 ^a,C^	147.83 ± 4.02 ^a,D^	141.20 ± 4.34 ^a,b,B^
Week 6	150.38 ± 3.72 ^a,C^	161.80 ± 5.26 ^a,B,C^	150.25 ± 4.01 ^a,C,D^	136.00 ± 4.69 ^b,B^
Week 8	172.67 ± 5.40 ^a,B^	177.00 ± 4.58 ^a,B^	166.20 ± 3.20 ^a,B,C^	130.43 ± 4.44 ^b,B^
Week 10	133.29 ± 3.19 ^c,C^	165.56 ± 2.91 ^a,B^	149.00 ± 5.80 ^b,C,D^	147.33 ± 4.76 ^b,B^
Mean	173.13 ± 5.18 ^a,b^	178.71 ± 2.93 ^a^	167.44 ± 2.59 ^b,c^	162.88 ± 4.50 ^c^

^1^ CON, control group; HFD, high-fat diet; HFD-NS5%, high-fat diet +5% Nelumbinis Semen powder; HFD-NS10%, high-fat diet +10% Nelumbinis Semen powder. Data expressed as mean ± SE. Values with a different ^a–c^ lowercase (^A–D^ uppercase) superscript in the same row (column) are significantly different at *p* < 0.05 by Duncan’s multiple range test. NS: not significant.

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
