# Peer review of "Anti-Obesity and Antidiabetic Effects of Nelumbinis Semen Powder in High-Fat Diet-Induced Obese C57BL/6 Mice"

_nutrients, 2020, doi:10.3390/nu12113576_

Round 1

Reviewer 1 Report

This study investigates the anti-diabetic and anti-obesity potential of Nelumbinis Semen extract, which comes from a traditional Korean medicinal plant. The authors added the compound to a high-fat diet at two doses and evaluated parameters such as organ and adipose tissue weights, glucose tolerance, insulin tolerance, and gene expression. While there was no significant reduction in body weight for the experimental animals, the authors found that several adipose tissues were significantly reduced in the lower-dose NS treatment. They also observed improvements in HOMA-IR scores, GLUT4 protein expression, and insulin tolerance, specifically with the lower dose NS treatment. 

While I appreciate the thoroughness of experimental design and the attempt to understand the mechanisms driving the expressed phenotype, there is one major flaw in the experimental design that cannot be ignored, and that is the improper use of a control diet. The authors indicated that they used a purified ingredient high-fat diet from Teklad and purchased matched diets containing the compound at different doses. Rather than purchasing the appropriate low-fat purified ingredient based diet, they used a grain based ('chow') diet as the low-fat control (see doi: 10.1186/s12986-018-0243-5). Given that these diets are wildly different in terms of nutrient composition and particularly the fiber content, it is difficult to give credence to the results. The authors will need to address this experimental limitation in detail. 

Aside from this major concern, here are some minor revisions/concerns that I would like to suggest:

  1. The authors spend a substantial amount of time talking about obesity and diabetes in the introduction, yet the section about NS and the experimental design is very brief. The authors mention that other studies have examined how this compound may impact obesity/diabetes onset, and therefore it is unclear how this study is novel and unique compared to previous work. 
  2. You may wish to consider showing the data from the analyses from the NS powder in the methods, so that the reader understands early on what types of compounds/doses are present in this material.
  3. I would suggest adding the caloric profile of the diets to Table 1. High-fat diets can range in terms of fat content, is this a moderately high-fat diet (45 kcal% fat) or a severe high-fat diet (60 kcal% fat). 
  4. How did the authors decided on the dose of 5 or 10% NS powder and what would be the equivalent dose in a human? Is it a physiologically relevant dose? 
  5. I think the authors put too much emphasis on the NS being an 'anti-obesity' compound. The experimental animals did not demonstrate significant weight reduction compared to the HF control. Yes, some adipose tissue weights were reduced in some cases, but measurements of body composition would be more indicative of the control of adiposity. Body composition was not measured in this study, so it seems to be a bit of a stretch to say that NS prevents obesity based on the current results. The authors should acknowledge this in their discussion, which currently contains no discussion of limitations at all. 
  6. I appreciate the authors' attempt to compare the results of their study with other published work. It would be helpful if more details were provided (study length, dose, etc) for the comparative studies so the reader understands how similar or different the study designs are compared to the current experiment. This would also help the reader understand if and why these results are novel compared to what is already published.

Author Response

Dear reviewer,

We thank for your time and expertise in reviewing the manuscript entitled "Anti-Obesity and Antidiabetic Effects of Nelumbinis Semen Powder in High-Fat Diet-Induced Obese C57BL/6 Mice."

Revisions were made according to the reviewer’s comments in the text and are highlighted in red color. Detailed explanations about the changes we made are attached. Please see the attachment. 

We are confident that we have adequately addressed the reviewers’ concerns. 

We hope you agree with the importance of our findings and now find the manuscript suitable for publication.

Sincerely,

Bog-Hieu Lee, Ph.D.

Department of Food and Nutrition, Chung-Ang University, Gyeonggi-do 17546, Korea

Tel.: +82-31-670-3276

lbheelb@cau.ac.kr

Reviewer 2 Report

This paper examines the anti-obesity, anti-diabetic, blood glucose regulation, and insulin resistance characteristics of Nelumbinis Semen (NS), the seed of Nelumbo nucifera Gaertn, and is valuable. The following items should be added or modified so that researchers who intend to utilize this paper can understand the contents of this research accurately and refer to it for confirmation of reproducibility.

  1. Regarding NS, I think that the ingredients may change depending on the variety, harvest timing, etc. In this study, it is said that NS extract was simply prepared and examined, but where is the origin (Vietnam?). What are the varieties such as subtypes,which was used for preparation of the NS powder products. Please give us a little more detail about what it is.

  1. About Table.1. The contents are displayed in %. Is this % by weight? In order to adjust the same HFD, please explain that it can be reproduced as a notation such as how many grams of each component are mixed. Also, NS powder is also displayed in%. Is this a ratio as a dry weight? If not, it is necessary to specify the amount of water, etc., and use a notation that indicates the amount that can be reproduced.

  1. Is it the mouse 12 weeks after the test feeding that the blood and tissue were collected? There is a description that the mice were bred until the 12th week, but I think it would be good if there is a description about the test breeding period of the dissected mice in sections 2.4 and 2.5 on page 5.

  1. A detailed description of the mouse used for IPGTT is required. Probably because it was conducted as an independent test system, it is necessary to explain the situation of the mice for this experiment, such as how long the test feeding for this mouse is.

  1. About Table 4. Is there any reason why the daily food intake of HFD is lower than others?

  1. I think Table 4 and Figure 1 show the same thing, so why not unify them?

  1. To explain the action at the molecular level, if possible, consider showing the expression of each molecule in adipose tissue as well as the liver. For example, I think that comparison of the expression state of GLUT4 in adipose tissue is better for explaining the pathway. Also, if you show the expression status not only at the protein level but also at the mRNA level, I think it will be a double check.

    This paper examines the anti-obesity, anti-diabetic, blood glucose regulation, and insulin resistance characteristics of Nelumbinis Semen (NS), the seed of Nelumbo nucifera Gaertn, and is valuable. The following items should be added or modified so that researchers who intend to utilize this paper can understand the contents of this research accurately and refer to it for confirmation of reproducibility.

    1. Regarding NS, I think that the ingredients may change depending on the variety, harvest timing, etc. In this study, it is said that NS extract was simply prepared and examined, but where is the origin (Vietnam?). What are the varieties such as subtypes,which was used for preparation of the NS powder products. Please give us a little more detail about what it is.

    1. About Table.1. The contents are displayed in %. Is this % by weight? In order to adjust the same HFD, please explain that it can be reproduced as a notation such as how many grams of each component are mixed. Also, NS powder is also displayed in%. Is this a ratio as a dry weight? If not, it is necessary to specify the amount of water, etc., and use a notation that indicates the amount that can be reproduced.

    1. Is it the mouse 12 weeks after the test feeding that the blood and tissue were collected? There is a description that the mice were bred until the 12th week, but I think it would be good if there is a description about the test breeding period of the dissected mice in sections 2.4 and 2.5 on page 5.

    1. A detailed description of the mouse used for IPGTT is required. Probably because it was conducted as an independent test system, it is necessary to explain the situation of the mice for this experiment, such as how long the test feeding for this mouse is.

    1. About Table 4. Is there any reason why the daily food intake of HFD is lower than others?

    1. I think Table 4 and Figure 1 show the same thing, so why not unify them?

    1. To explain the action at the molecular level, if possible, consider showing the expression of each molecule in adipose tissue as well as the liver. For example, I think that comparison of the expression state of GLUT4 in adipose tissue is better for explaining the pathway. Also, if you show the expression status not only at the protein level but also at the mRNA level, I think it will be a double check.

Author Response

(The authors gave the same response as above.)

Reviewer 3 Report

The authors described in this manuscript the anti-obesity and anti-diabetic effects of nelumbinis semen powder in C57BL/6 mice fed a high-fat diet.

Major comments:

1. Nelumbinis semen extract has been used in the literature mainly as an antioxidant and cardiovascular symptoms, however, the authors wanted to test its obesogenic and diabetic effects. What is the rationale?

2. This study is merely descriptive and shows some of the biochemical and histological parameters and is insufficiency many molecular details. The work would have had a good impact if the authors had described more mechanism of action and signaling pathway.

3. The reviewer thinks that the most important transcription factor of adipogenesis and glucose metabolism is C/EBPß and/or PPAR∂. The authors should examine the mRNA and protein expression of these molecules and discuss the side of action of nelumbinis semen powder more precisely.

4. In order to understand better the effect of nelumbinis semen powder inhibiting differentiation, the authors need to show the effects with a known compound as a positive control.

5. The materials and methods are not very descriptive and do not clearly indicate how the experiments were conducted. In particular, information of nelumbinis semen powder is very insufficiency. Even if authors have purchased an extract, it is very important not only the extraction information and date of manufacture, but also the information on the substance and the description of the main compound in the extract.

Author Response

(The authors gave the same response as above.)

Round 2

Reviewer 1 Report

I appreciate the authors' diligence in making improvements to the manuscript. I find it to be much improved compared to the original version. 

Reviewer 3 Report

This revised manuscript has supported the results and conclusion.